# Molecular dynamics simulations of human cohesin subunits identify DNA binding sites and their potential roles in DNA loop extrusion

**Chenyang Gu**[ID], **Shoji Takada, Giovanni B. Brandani**\*, **Tsuyoshi Terakawa**[ID]\*

Department of Biophysics, Graduate School of Science, Kyoto University, Kyoto, Japan

\* terakawa@biophys.kyoto-u.ac.jp (TT); brandani.giovannibruno.5v@kyoto-u.ac.jp (GBB)

## Abstract

The SMC complex cohesin mediates interphase chromatin structural formation in eukaryotic cells through DNA loop extrusion. Here, we sought to investigate its mechanism using molecular dynamics simulations. To achieve this, we first constructed the amino-acid-residue-resolution structural models of the cohesin subunits, SMC1, SMC3, STAG1, and NIPBL. By simulating these subunits with double-stranded DNA molecules, we predicted DNA binding patches on each subunit and quantified the affinities of these patches to DNA using their dissociation rate constants as a proxy. Then, we constructed the structural model of the whole cohesin complex and mapped the predicted high-affinity DNA binding patches on the structure. From the spatial relations of the predicted patches, we identified that multiple patches on the SMC1, SMC3, STAG1, and NIPBL subunits form a DNA clamping patch group. The simulations of the whole complex with double-stranded DNA molecules suggest that this patch group facilitates DNA bending and helps capture a DNA segment in the cohesin ring formed by the SMC1 and SMC3 subunits. In previous studies, these have been identified as critical steps in DNA loop extrusion. Therefore, this study provides experimentally testable predictions of DNA binding sites implicated in previously proposed DNA loop extrusion mechanisms and highlights the essential roles of the accessory subunits STAG1 and NIPBL in the mechanism.

## Author summary

The study explores the molecular dynamics of human cohesin, a protein complex crucial for the structural organization of chromosomes in eukaryotic cells. Using molecular dynamics simulations, we identified specific DNA-binding sites within cohesin's subunits, including SMC1, SMC3, STAG1, and NIPBL. These sites play a pivotal role in cohesin's ability to mediate DNA loop extrusion—a process essential for genome organization and gene regulation. By constructing structural models of the cohesin complex, we uncovered cooperative interactions among DNA-binding patches. These interactions stabilize DNA and facilitate its bending, vital for forming chromatin loops. The study also highlights the

**Data availability statement:** The data that support the findings of this study are publicly available in The Biological Structure Model Archive (BSM-Arc) at https://bsma.pdbj.org/entry/70

**Funding:** This work was supported by the Research Incentive Grant provided by "Support for Pioneering Graduate Students" presented by the Kyoto University Graduate Division (to C.G), the Japan Society for the Promotion of Science KAKENHI grant (20H0593; to ST, 21H02441; to S.T., 24K01991; S.T., 20K06587; to G.B.B.), the MEXT grant JPMXP1020230119 as "Program for Promoting Researches on the Supercomputer Fugaku" (to S.T.), the Japan Society for the Promotion of Science Grant-in-Aid for Transformative Research Areas (24H00883; to T.T.), the grant from the Kyoto University Foundation (to T.T.), the grant from the Takeda Science Foundation (to T.T.), the grant from the Shimazu Science Foundation (to T.T.), and the grant from the Inamori Foundation (to T.T.). The funders had no role in study design, data collection and analysis, decision to publish, or preparation of the manuscript.

**Competing interests:** The authors have declared that no competing interests exist.

indispensable roles of the accessory proteins STAG1 and NIPBL, whose absence may disrupt DNA binding. This work provides new insights into the mechanisms underpinning DNA loop extrusion, validating previously proposed models and suggesting novel sites for experimental investigation. These findings deepen our understanding of cohesin's role in genome architecture and open avenues for further studies into chromatin dynamics and gene expression regulation.

## Introduction

Cohesin, a structural maintenance of chromosomes (SMC) complex, is a molecular motor that mediates chromosome structural regulation in eukaryotic cells. In addition to its role in sister chromatid cohesion, cohesin is involved in the formation of topologically associating domains (TAD) in interphase chromatin [1]: self-interacting genomic regions insulated from the outside [2]. TAD formation plays a critical role in gene expression regulation by facilitating or preventing interactions between promoters and enhancers [3,4]. As a mechanism for the TAD formation by cohesin, the DNA loop extrusion mechanism [5] was previously proposed. During loop extrusion, cohesin captures a small DNA segment inside its ring-like protein structure and expands it by extrusion until it encounters insulator elements such as CTCF. The mechanism was proposed based on the observation that cohesin and pairs of convergent CTCF binding sites are often co-localized at the boundaries of TADs [6]. Single-molecule imaging studies confirmed that cohesin, as a molecular motor, can extrude a DNA loop using adenosine triphosphate (ATP) hydrolysis energy [7,8].

The human cohesin complex forms a hetero-pentamer (Fig 1). The SMC1 and SMC3 subunits dimerize at their hinge domain and can also engage at their ATPase head domain, depending on the bound nucleotides. The N- and C-terminal regions of the intrinsically disordered RAD21 subunit attach to the SMC3 and SMC1 head domains, respectively. The two accessory subunits, STAG1 and NIPBL, bind to RAD21 and are essential for motor activity, though their exact roles still need to be clarified [7,8]. Cohesin changes its conformation or DNA-binding state throughout the ATP hydrolysis cycle [9–11]. SMC1/3 heads engage upon ATP binding and disengage after ATP hydrolysis, while NIPBL associates with RAD21 upon the head engagement and dissociates after ATP hydrolysis. Upon ATP binding to SMC1/3, about 50 nm long anti-parallel coiled-coil arms that connect their heads and hinges are curved to take an open "O" conformation (Fig 1A). The entire cohesin complex transitions to an

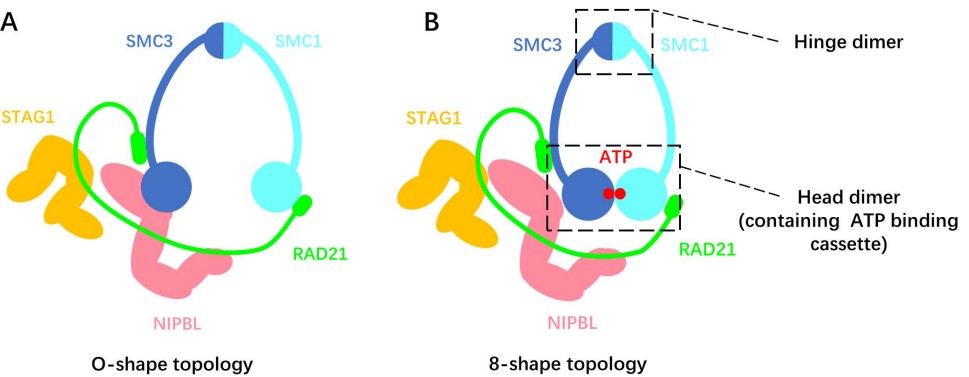

**Fig 1. An illustration of the human cohesin complex in (A) O-shape and (B) 8-shape topology.**

"8"-like conformation (Fig 1B) upon the head engagement, with the whole complex forming two distinct compartments where DNA could potentially be entrapped: one formed by the coiled coils from the hinge up to the engaged heads, and another formed by RAD21. Upon head disengagement, these two compartments merge into one larger compartment. These conformational changes have been suggested to drive DNA loop extrusion [12,13].

Previous studies have adopted numerous experiments, including high-speed atomic force microscopy (HS-AFM) [9,10], cryogenic electron microscopy (cryo-EM) [14,15], and single-molecule fluorescence imaging [7,8], to investigate the interaction between SMC complexes and DNA. Despite their success, HS-AFM and fluorescence imaging lack sufficient resolution to identify the precise DNA-binding sites. Cryo-EM requires averaging over many structural snapshots to obtain high-resolution structures, potentially missing transient interactions between DNA and other parts of the complex. For example, mutating the basic amino acids on the surface of SMC1/3 hinge dimer facing the inside of the ring compartment formed by coiled coils or on the C-terminus of STAG1's HEAT repeats domain abrogate loop extrusion [9], but DNA binding at these sites was not observed by cryo-EM. Therefore, it is reasonable to assume that some critical DNA binding sites involved in loop extrusion have not been resolved in the previous works. So far, multiple models for the loop extrusion mechanism have been proposed, such as the DNA segment capture model [13], the scrunching model [5], the swing and clamp model [9], and the hold and feed model [16]. They all assume that a DNA-binding site captures a DNA segment and hands it over to another throughout the ATP hydrolysis cycle. Therefore, identifying new DNA binding sites is essential for validating and refining these models or proposing new ones. To date, Higashi et al. [17] and Nomidis et al. [18] have performed 3D simulations using coarse-grained models that represent cohesin or SMC complexes in general, respectively, using a small number of rigid segments connected by flexible joints. Both studies have obtained valuable insights into the DNA loop extrusion mechanism. However, the models omitted or highly coarse-grained the interactions between protein and DNA. Prediction of DNA binding sites should help refine such coarser models.

In this study, we performed molecular dynamics (MD) simulations of human cohesin subunits with DNA to comprehensively predict and rank the strength of the DNA binding sites on these proteins. We also modeled the whole cohesin complex and mapped the predicted binding sites to infer the potential pathways of DNA handover in the context of DNA loop extrusion.

## Result

### Residue-resolution modeling of human cohesin subunits

In this work, we predicted DNA binding sites in cohesin subunits and quantified their strength by measuring the rates of DNA unbinding from the simulations. For this computationally demanding task, we used the AICG2+ protein model, where one particle represents one amino acid, and the 3SPN.2C DNA model [19,20], where three particles represent one nucleotide. Debye-Hückel electrostatic interactions and excluded volume interactions were imposed between protein and DNA. This model allows a speedup of several orders of magnitude compared to all-atom models while retaining key molecular details and predictive capacity, having been already applied to a variety of systems, including *S. cerevisiae* condensin [21], human PRC2 complex [22], and DNA binding proteins HoxD9, Sap1, and Skn1 [23], to predict their DNA binding sites successfully.

We built residue-resolution coarse-grained structural models of essential subunits of human cohesin. The initiation and progression of DNA loop extrusion require the SMC1, SMC3, RAD21, and STAG1 subunits and the C-terminal region of the NIPBL subunit in

addition to ATP and DNA [7,8]. We chose these five subunits as our target. The cryo-EM structure of the human cohesin-DNA complex (PDB ID: 6WG3) [14] shows the gripping state in which these subunits load on DNA with the heads of SMC1 and SMC3 bound to ATP. We used the partial structures from this complex to build the residue-resolution structural model of each subunit.

The cryo-EM structure has missing residues: the parts of the coiled-coil arms in SMC1 and SMC3 and the intrinsically disordered regions (IDR) flanking the HEAT repeats of STAG1 and NIPBL. Therefore, we built residue-resolution structural models of the following eight constructs (Fig 2A to 2H): (A) the SMC1 head and (B) the SMC3 head with the

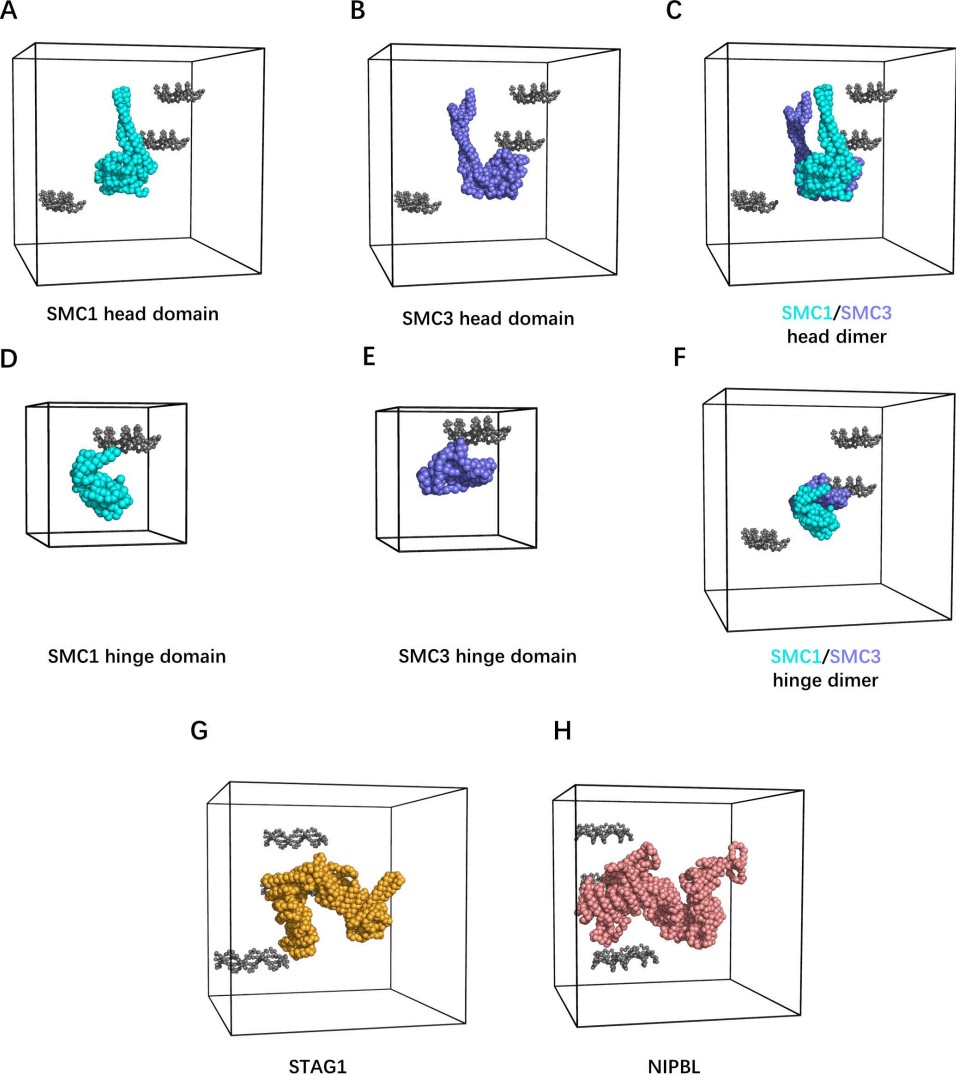

**Fig 2. Residue-resolution modelling of human cohesin subunits and DNA binding simulation setup.** (A to H) Simulation setup of DNA binding assays for cohesin subunits. Gray-colored beads indicate the DNA model, and the black cube indicates the periodic simulation boundary. Cyan, blue, orange, and pink colored beads indicate SMC1 head domain with an emanating partial coiled-coil arm in (A), SMC3 head domain with an emanating partial coiled-coil arm in (B), SMC1/3 head dimer with emanating partial coiled-coil arms in (C), SMC1 hinge domain in (D), SMC3 hinge domain in (E), SMC1/3 hinge dimer in (F), the HEAT repeats domain of STAG1 in (G), and the HEAT repeats domain of NIPBL in (H).

cryo-EM solved part of the coiled-coil arms, (C) the dimer of A and B, (D) the SMC1 hinge, (E) the SMC3 hinge, (F) the dimer of D and E, (G) the HEAT repeats of STAG1, and (H) NIPBL. These include monomeric and heterodimeric domains to investigate the cooperatively created binding site. We omitted the IDR of STAG1 and NIPBL from the models because of their low sequence conservation, suggesting a minor contribution to loop extrusion (S1 Fig). Consistent with this, the N-terminal IDR of NIPBL has been proven unessential for *in vitro* DNA loop extrusion [8], while the C-terminal IDR of STAG1 and NIPBL plays a role in loading to DNA [10].

We then prepared residue-resolution models of short DNA segments. In the cryo-EM structure [14], cohesin forms a complex with 72 base pairs (bp) of double-stranded DNA. However, the experiment only captures 17 bp of DNA bound to cohesin and does not identify the sequence of the captured DNA. To study the non-specific DNA binding of the cohesin subunits, we used a randomly generated 20 bp DNA sequence with the same GC content as the DNA used in the cryo-EM experiment [14] (Methods). When multiple DNA segments were introduced, we used copies of the same DNA sequence.

The residue-resolution structural models of each protein subunit and DNA segments were constructed using CafeMol [24] (Fig 2). In the AICG2+ protein model and forcefield [25], one particle at the α-carbon position represents one amino acid residue. The force field stabilizes the native protein conformation, with the equilibrium bond lengths, bond angles, dihedral angles, and non-bonded native contact lengths extracted from reference all-atom structures. In the 3SPN.2C DNA model [20], three particles at the centers of mass of the base, sugar, and phosphate units represent one nucleotide. Proteins and DNA interact via Debye-Hückel electrostatics, which may cause attraction/repulsion, and excluded volume, which is purely repulsive. Partial charges on the protein particles were arranged using the RESPAC algorithm [26] to reproduce the electrostatic potential around the reference all-atom structures.

We performed residue-resolution coarse-grained MD simulations to identify DNA-binding sites on cohesin subunits and quantify the affinities (Fig 2A to 2H). Protein structural models SMC1 head, SMC3 head, SMC1/3 head dimer, SMC1/3 hinge dimer, NIPBL, and STAG1 are each placed inside a small cubic box with periodic boundaries, together with three copies of 20 bp DNA fragments randomly placed around the protein (see the Materials and methods section for more details). Including multiple DNAs increases the efficiency of the binding site search due to the high concentration and the fact that two DNA segments can continue exploring even if one is trapped at a particular site. SMC1 hinge and SMC3 hinge are each placed inside a periodic box with only one copy of the 20 bp DNA since the small size of these subunits naturally facilitates the identification of the binding sites.

## Molecular dynamics simulations predicted DNA binding patches in cohesin

We performed residue-resolution MD simulations on systems A to H as described above (S1 Movie). In all systems (using system A, SMC1 head domain with short emanating coiled coils, as an example in Fig 3), the simulation trajectories showed DNA segments repeatedly associating with and dissociating from the protein surface residues (Fig 3A). We considered a DNA segment in contact with an amino acid when the smallest distance from any particle of the DNA segment to the amino acid particle is within a given threshold (see Methods for details). Based on this definition, we identified the amino acids in contact with each DNA segment as a function of time (Fig 3B). Surface amino acids that are spatially close to each other tend to encounter the same DNA segment at a given time, and DNA-contacting probabilities mapped on the structure show that amino acid particles with high contact probability cluster at several

spatially localized regions (S2 Fig). Hereafter, we refer to these spatial regions as binding patches. The clear pattern of simultaneous binding with DNA within each patch allowed us to quantitatively identify patches with a clustering analysis using the Jaccard index describing the similarity in DNA contact patterns between two amino acid particles (Methods, S3 Fig). The identified binding patches are mapped on the structure (Figs 3C and S4).

To evaluate the DNA-binding affinity of each patch, we calculated their dissociation rate constants from the simulation trajectories. We reasonably assumed that DNA binding is diffusion-limited, implying a similar association rate constant for all patches, which allows us to use the dissociation rate constant as a proxy for relative affinity. To estimate the dissociation

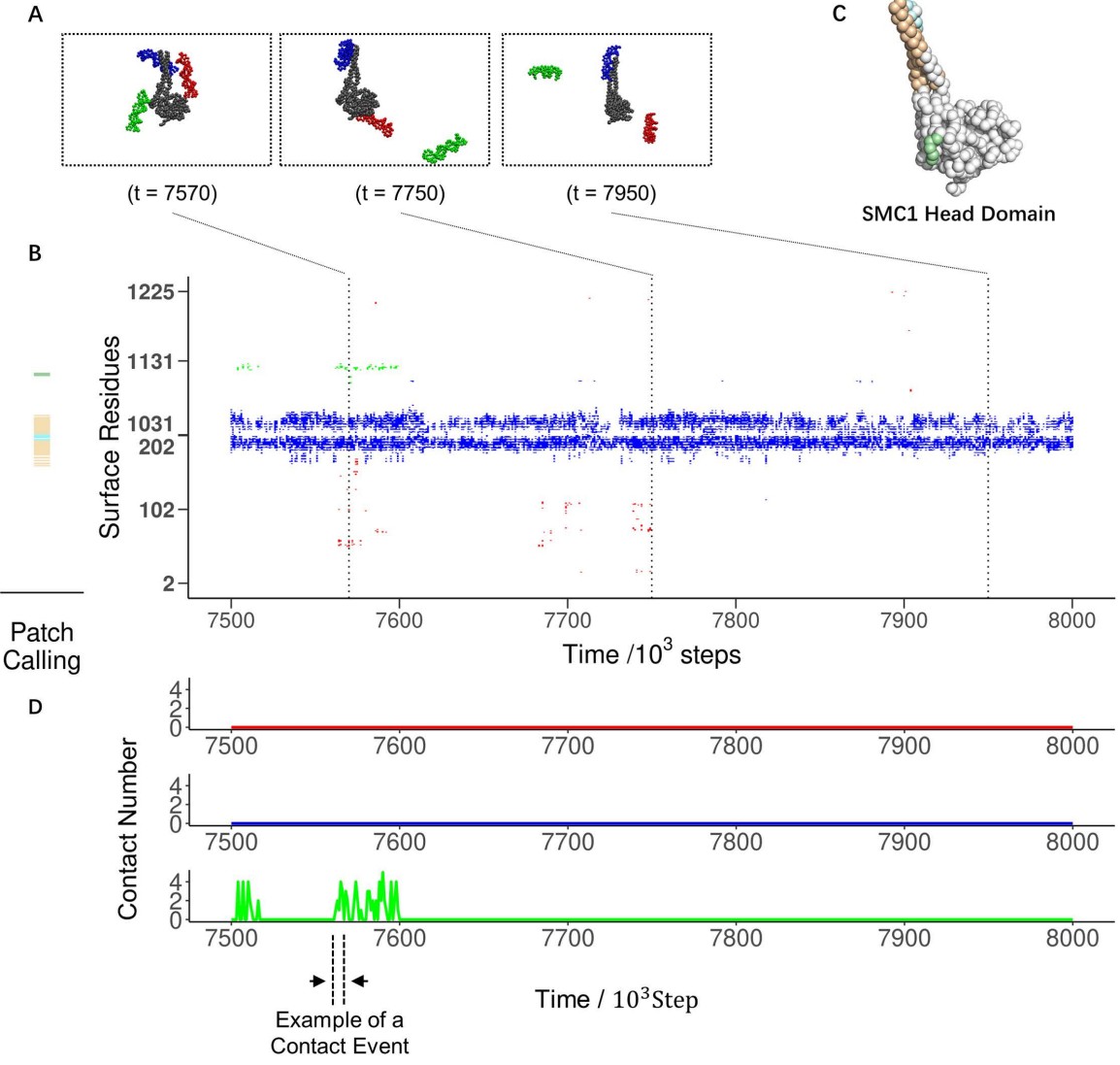

**Fig 3. Analysis of DNA binding simulation assay trajectories.** (A) Representative snapshots of of the SMC1 head domain with DNA (System A in Fig 2) taken from the simulation trajectory shown in (B). The numbers indicate simulation time step $t$ in the unit of 1000 steps. One (right), two (center), or three (left) DNA segments are in contact with SMC1 surface residues in each snapshot. (B left) DNA binding patches identified on the SMC1 head domain using all 20 trajectories. (B right) Time series of IDs of amino acid particles in contact with DNA. The colors indicate which DNA (red, blue, or green) in (A) is in contact with the protein. (C) Binding patches presented in (B left) mapped on the structure of the SMC1 head domain. (D) Time series of the number of DNA-contacting residues in the light green patch in (C). An example of a single contact event is shown with dotted lines and arrows.

rates, we measured the time intervals in which DNA stays in contact with each patch [Fig 3D, using patch A2 as an example. A2 is light green in Fig 3B left panel and Fig 3C. Throughout this paper, we systematically name patches by combining the name of the system (here A) and the affinity rank (here second).] and used these to compute their survival probabilities (S5 Fig, Methods), representing the fraction of DNA molecules staying on the patch after a particular duration. Fitting exponential functions to the curve provided us with the dissociation rate constants. The sum of two exponential functions fitted the curves markedly better than one exponential function for all the patches (S5 Fig), suggesting the two interaction modes: the transient and stable modes. Here, we used the dissociation rate constant of the stable mode (the lower one) as a proxy for relative affinity to ignore the random encounter.

The analysis identified multiple DNA binding patches on each subunit and its dimeric form (Table 1). We focused on the patches that associate more strongly with DNA than the patch on the top surface of the heads of SMC1/3 heterodimer (Fig 4A), as those most likely to play a significant role in loop extrusion, as DNA is clamped on this surface in the cryo-EM structure of the gripping state [14]. Therefore, we predicted the DNA binding patches with a dissociation rate constant lower than $2.0 \times 10^{-5}$ timesteps$^{-1}$ (Fig 4B to 4E) as the potential DNA binding patches. Table 1 lists these binding patches, their dissociation rate constant, key positively charged residues, and eventual remarks on the patches.

In the SMC1 head domain (A) and head dimer systems (C), we observed strong patches A1 and C1 (Table 1), corresponding to the same group of amino acids on the coiled-coil part connected to the SMC1 head domain. These patches were always bound to at least one DNA segment throughout the simulation trajectories. This means the dissociation of one DNA segment from these patches resulted from another DNA segment competing for the same binding site (S6 Fig). Although association duration for these patches still follows exponential distributions, the dissociation rate constants calculated under this condition have different physical meanings than other patches and should not be compared directly. However, the value we obtained in this way gives us a lower bound on the ideal dissociation rate constants. Since the values for these patches are the smallest among all the identified patches, we know that these patches have strong interaction with DNA and should be considered essential candidates as patches involved in DNA loop extrusion.

In each system, multiple DNA binding patches were identified (S4 Fig and Table 1). In the dimer systems SMC1/3 head dimer (C) and SMC1/3 hinge dimer (F), dimerization caused some surface amino acids to be hidden, altering the DNA binding patches in the corresponding monomers. The dimerization partly hid patch B1 (side surface of SMC3 head domain), while the exposed part formed a new patch C3 (top surface of SMC1/3 head dimer) with some amino acids on the SMC1 head domain (S4A to S4C Fig). Dimerization of the SMC1 and SMC3 hinge domains unified the patches D2, E1, E2, E3, and E4. These patches collectively formed a patch F1, which contained all the surface residues of the SMC3 hinge domain and some amino acids from the SMC1 hinge domain (S4D to S4F Fig). SMC1 and SMC3 head domains engage and disengage during the ATP hydrolysis cycle, powering DNA loop extrusion, making dynamic changes in DNA binding patches potentially relevant for the loop extrusion mechanism. Hinge opening is also hypothesized to play a role in the loading process of cohesin [27]. Therefore, binding patches on SMC1 head domain (A), SMC3 head domain (B), SMC1 hinge domain (D), SMC3 hinge domain (E), SMC1/3 head dimer (C), and SMC1/3 hinge dimer (F) should all be investigated when studying the loop extrusion mechanism.

Consistent with cohesin's role of mediating loop structures across the genome, no sequence-specificity has been reported on cohesin's subunits SMC1, SMC3, RAD21, STAG1, or NIPBL. Nevertheless, to address a potential bias introduced when using a specific DNA sequence, we performed DNA binding simulations on STAG1's structured domain (G)

with various DNA sequences differing in CG content (20%, 45% and 80%, see Methods for the DNA sequence used in corresponding simulations). DNA dissociation rate constants calculated from these simulations gave similar binding strength for the 4 strongest patches predicted on STAG1 (S7 Fig). STAG1 contains multiple DNA binding patches, including both strong patches (dissociation rate constant below $2.0 \times 10^{-5}$ timestep$^{-1}$) and weak patches (dissociation rate constant above $2.0 \times 10^{-5}$ timestep$^{-1}$), and does not contain the patch that cannot be precisely quantified (patch A1, described above"). We consider the DNA binding patches on STAG1 a good sample of all the DNA binding patches identified in this study to eliminate the possibility of DNA sequence-induced bias.

**Table 1. DNA Binding patches with a binding affinity strong enough to participate in DNA handover during translocation or loop extrusion.**

| Sub-unit/domain | Patch name | $K_{off} \pm sd(k_{off})$ | Positive charged residues | Remarks |
|---|---|---|---|---|
| SMC1 head | A1 | $1.2 \times 10^{-7} \pm 1 \times 10^{-8}$ | 170, 171, 187, 188, 189, 190, 196, 197, 200, 1049, 1050, 1051, 1053, 1054, 1056, 1064 | |
| SMC3 head | B1 | $1.25 \times 10^{-5} \pm 5 \times 10^{-7}$ | 33, 55, 57, 143, | |
| | B2 | $1.59 \times 10^{-5} \pm 4 \times 10^{-7}$ | 72, 105, 106, 113, 185, 997, 1003 | |
| SMC1/3 head dimer | C1 | $1.20 \times 10^{-7} \pm 9 \times 10^{-9}$ | SMC1: 170, 171, 172, 173, 187, 188, 189, 190, 196, 197, 200, 1035, 1049, 1050, 1051, 1053, 1054, 1056 | ① |
| | C2 | $1.6 \times 10^{-5} \pm 2 \times 10^{-6}$ | SMC1: 59 | ② |
| | C3 | $1.60 \times 10^{-5} \pm 4 \times 10^{-8}$ | SMC3: 55, 57, 72, 105, 106, 113, 114 | ③ |
| | C4 | $1.62 \times 10^{-5} \pm 7 \times 10^{-7}$ | SMC3: 185, 188, 997, 999, 1002, 1003, | |
| SMC1/3 hinge dimer | F1 | $2.19 \times 10^{-6} \pm 9 \times 10^{-8}$ | SMC1: 491, 496, 499, 500, 554, 561, 564; SMC3: 493, 498, 503, 514, 518, 520, 521, 522, 527, 533, 557, 561, 571, 578, 592, 596, 612, 614, 618, 621, 624, 625, 629, 634, 644, 660, 661, 672, 673, 675, 680, 683 | |
| STAG1 | G1 | $5.6 \times 10^{-6} \pm 3 \times 10^{-7}$ | 172, 173, 175, 352, 549, 550 | |
| | G2 | $7.2 \times 10^{-6} \pm 4 \times 10^{-7}$ | 718, 721, 759, 766, 767, 770, | |
| | G3 | $9.7 \times 10^{-6} \pm 4 \times 10^{-7}$ | 385, 910, 912, 957, 958, 1013, 1016 | |
| NIPBL | H1 | $3.4 \times 10^{-7} \pm 2 \times 10^{-8}$ | 1235, 1236, 1247, 1294, 1297, 1369, 1371, 1372, 1374, 1378, 1379, 1381, 1389 | ④ |
| | H2 | $9.3 \times 10^{-7} \pm 5 \times 10^{-7}$ | 1552, 1599, 1789, 1791, 1794, 1828, 1865, 1866, 1867, 1870, 1873, 1904, 1912 | ⑤ |
| | H3 | $5.1 \times 10^{-6} \pm 3 \times 10^{-7}$ | 2104, 2112, 2119, 2162, | |
| | H4 | $7.0 \times 10^{-6} \pm 1 \times 10^{-6}$ | 1779, 2563, 2574, 2578, 2583 | |
| | H5 | $1.0 \times 10^{-5} \pm 2 \times 10^{-6}$ | 1965, 1968, 1969 | |

Remarks:

① Same position as patch A1.

② In contact with DNA in the gripping state.

③ In contact with DNA in the gripping state. Head engagement results in some residues from patch B1 to be hidden, the rest form a new patch C3 with residues from patch B2.

④ In the gripping state, this patch is covered due to NIPBL-STAG1 binding.

⑤ In contact with DNA in the gripping state.

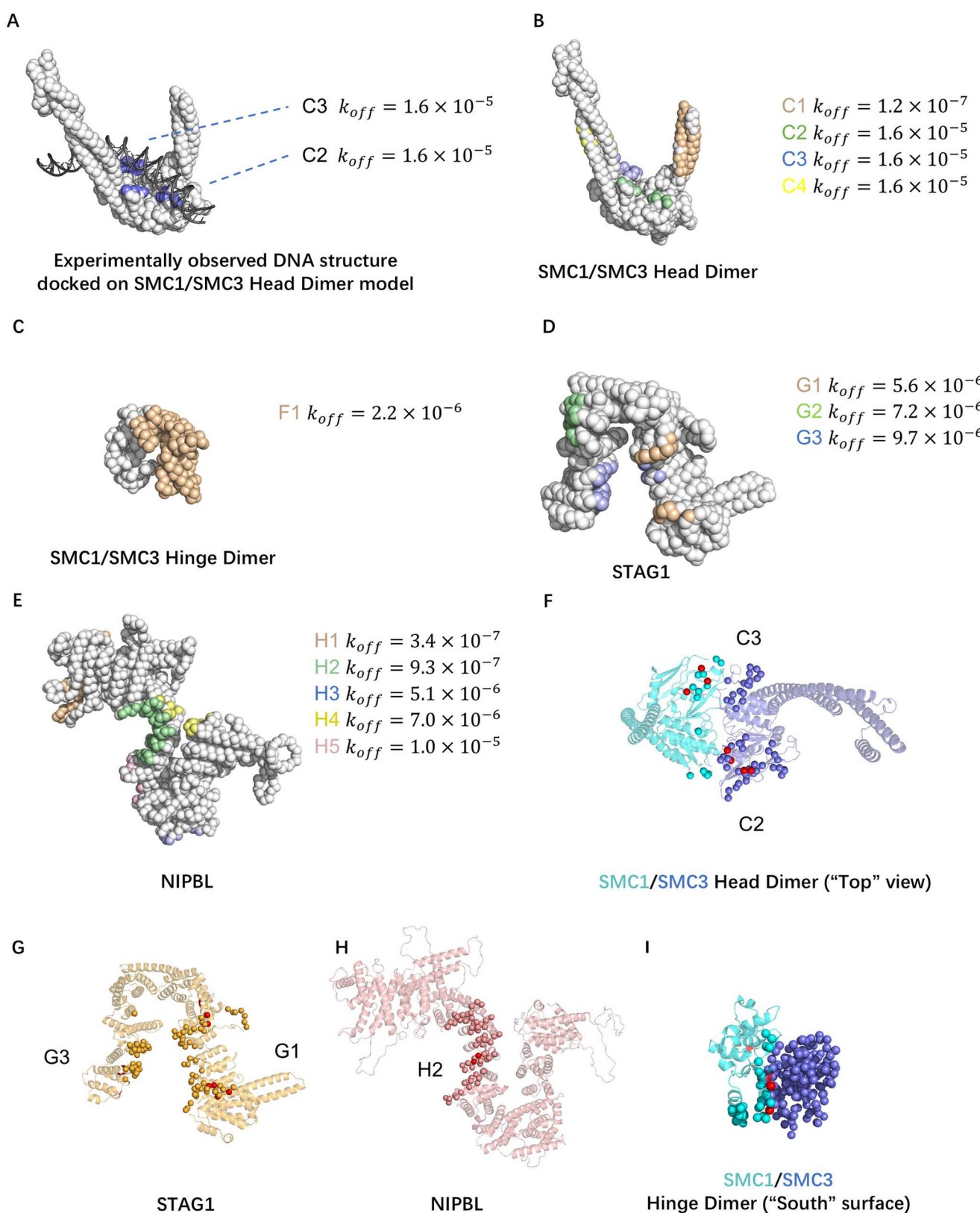

**Fig 4. Prediction and validation of strong DNA binding patches.** (A) The positions of DNA binding patches on the top surface of the SMC1/3 head dimer. The cryo-EM DNA structure (PDB ID: 6WG3) is superimposed on the coarse-grained model of the SMC1/3 head dimer. (B to E) DNA binding

patches with a binding affinity higher or comparable with patches in (A) on each subunit or their dimers. (F to I) Positions of the predicted strong patches (beads colored other than red) and experimentally identified critical amino acid residues (red beads) in the SMC1/3 head dimer, STAG1, NIPBL, and SMC1/3 Hinge dimer.

**Table 2. Predicted DNA binding patches that spatially or sequentially overlapped with the experimentally identified critical amino acids.**

| Simulation predicted patch | Experiment | Effect on loop extrusion |
|---|---|---|
| Head dimer patch #2 | SMC1 K52E, R57E, K59E, K62E | Abrogate |
|  | ΔSMC1 58-62 | Reduce |
| Head dimer patch #3 | SMC3 R57E, R61E, K105E, K106E | Abrogate |
|  | SMC3 R57E, R61E | Reduce |
| Hinge dimer patch F1 | SMC1 K551A, R554A, K561A, R644A, | Abrogate |
| STAG1 patch #1 | STAG1 K92E, K95E, K172E, K173E | Abrogate |
|  | STAG1 K555E, K558E, K564E | Abrogate |
| STAG1 patch #3 | STAG1 K969E, K971E, K1013E, K1016E | Abrogate |
| NIPBL patch #2 | NIPBL R1867A, R1870A | Abrogate |

## Electrostatics, in vivo mutagenesis, and molecular docking validated the predicted DNA binding patches

We now turn to the role of electrostatic interactions in the identified DNA binding patches on the proteins. In this study, the RESPAC algorithm determined charge arrangements on surface amino acid particles to reproduce the electrostatic potential around an all-atom protein structure calculated by the Adaptive Poisson-Boltzmann Solver [28]. The positive electrostatic potential surface area overlapped the predicted DNA binding patches in the simulation runs. Notably, when we replaced the RESPAC charges with simpler ±1 unit charges on basic and acidic amino acid particles, the DNA binding sites appeared more dispersed, thus weakening some key electrostatic features (e.g., the strong and concentrated positive charges at the 'neck' of NIPBL are recreated by RESPAC calibration, S8 Fig). Therefore, the charge arrangement calibration by the RESPAC algorithm assisted in predicting the DNA binding sites created by the cooperative contribution of charges.

The predicted DNA binding patches also overlapped with the amino acid residues already identified to play an essential role in DNA loop extrusion in the previous mutagenesis studies [9] (Fig 4F to 4I). Reference [9] studied nine mutation or deletion sites on SMC1/3, STAG1, or NIPBL, each mutation/deletion either abrogated or slowed down loop extrusion (Table 2). All nine mutated/deleted amino acid residue groups were in the predicted DNA binding sites, validating our simulation results (Fig 4F to 4I).

Our simulations also predicted new DNA binding patches, including residues that have yet to be considered in mutagenesis studies. For example, we identified a potential DNA binding site at the coiled-coil arm emanating from the SMC1 head. The crystal structure of Rad50 in complex with DNA supported this prediction [29]. Rad50, a DNA repair protein paralogous to cohesin, shares a similar amino acid sequence and molecular architecture. In the structure, DNA binds to the coiled-coil arm region emanating from the head, whose sequence is conserved to the corresponding predicted DNA binding site on cohesin and located at a similar location (S9A Fig). Therefore, our MD simulations predicted experimentally testable potential DNA binding patches.

We used the molecular docking tool HADDOCK [30] as an independent validation of our predicted DNA binding patches. We use one protein domain (SMC1 head domain [A] or NIPBL HEAT repeats domain[H]) and one 20 bp DNA fragment as input structures. Without

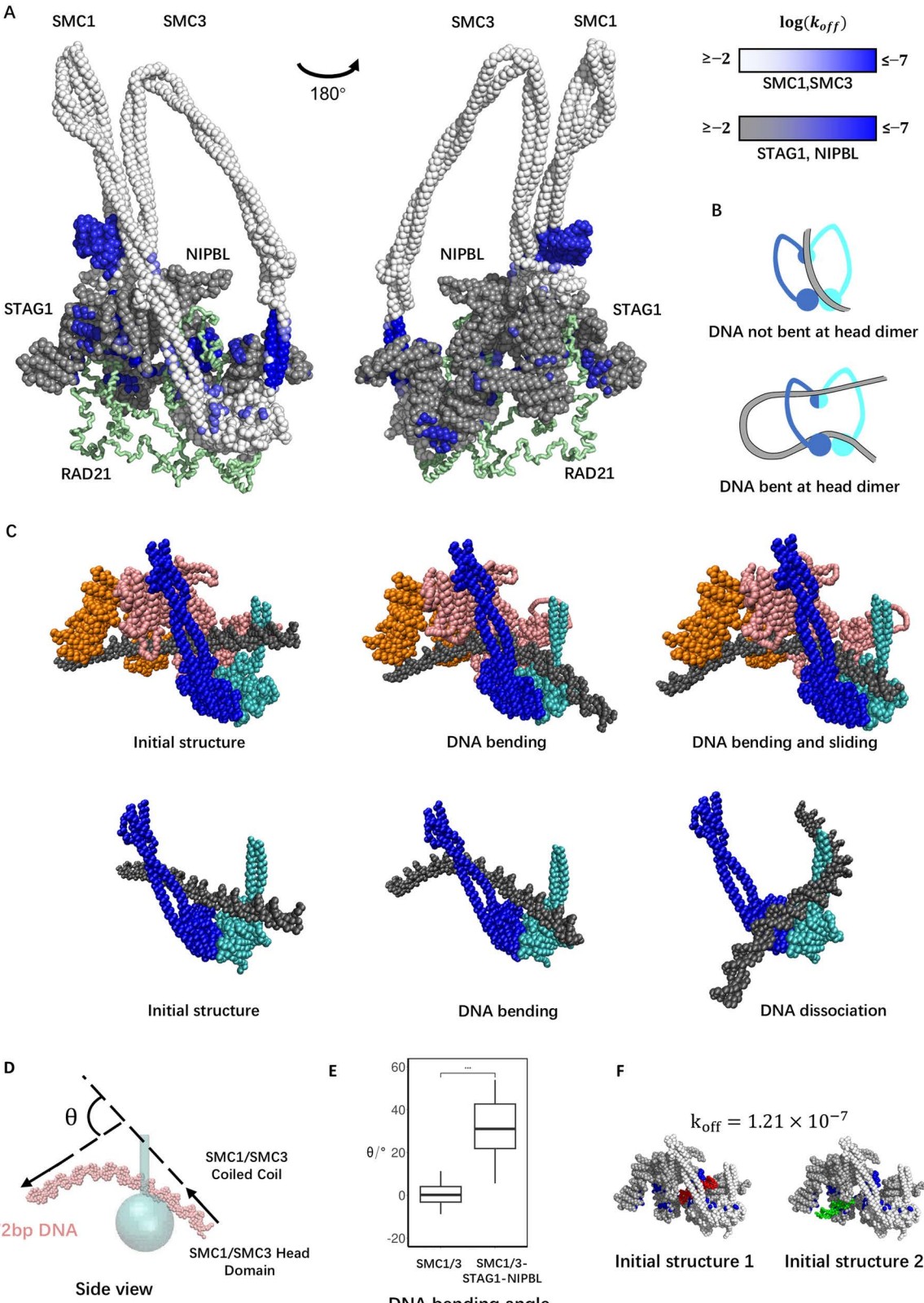

**Fig 5. Spatial relations of DNA binding patches in the whole cohesin complex and the cooperative effect of the patch group.** (A) The positions of all significant DNA binding patches identified in Fig 4 mapped on the structural model of the whole cohesin complex.

DNA binding affinity is indicated by blue intensity. The amino acid particles in SMC1 and SMC3 not identified as significant patches are colored in white, while those in STAG1 and NIPBL are colored in gray. Rad21 is visualized as a green tube. (B) Schematics of the DNA segment capture model. (C) The setup and representative snapshots of the simulations studying the effect of the DNA-gripping patch group. (C top) The simulation of the tetrameric complex containing the DNA-gripping patch group (SMC1/3 heads, STAG1, and NIPBL) with 72 bp dsDNA. (C bottom) The simulation of the SMC1/3 heads and the coiled coils emanating from them containing the partial DNA-gripping patch group (SMC1/3 heads). (D) An example snapshot of DNA projected on the plane cutting the cohesin ring (See S11A Fig for the definition of the plane) with the definition of the DNA bending angle $\theta$. (E) The distributions of $\theta$ calculated from the simulations of the dimeric (SMC1/3 heads) and the tetrameric (SMC1/3 heads, STAG1, and NIPBL) complexes. (***: p-value < 0.001) (F) The initial structures of the simulations of the tetrameric complex with 20 bp DNA. Surface amino acid particles forming the DNA-gripping patch group are colored blue. The DNA model is colored red and green.

specifying the interface regions, the DNA-protein docking interfaces predicted by HAD-DOCK overlap with the strongest DNA binding patches of each protein domain predicted by our MD simulations, A1 and H1 (Table 1 and S9B Fig). A1 and H1 are both novel DNA binding patches not studied in previous mutagenesis studies. When specifying positively charged surface residues selected from our other predicted patches as interface regions in HADDOCK input parameters, HADDOCK produced complex structures in which DNA docks to the corresponding patch (S9C Fig).

## Cooperative contributions of DNA binding sites may drive loop extrusion

We conducted structural modeling of the whole cohesin complex, including SMC1, SMC3, STAG1, NIPBL, and RAD21, to get insight into the relative positions of the DNA binding sites predicted above and to infer the possible handover pathways in DNA loop extrusion (Fig 5A).

As mentioned, in cryo-EM structure of the human cohesin-DNA complex, SMC1 and SMC3 have consecutive missing residues in their coiled-coil regions, and RAD21 has most of its IDR missing. The structures of these regions were predicted using AlphaFold Colab [31]. Then, we sought to build a residue-resolution structural model of the whole human cohesin complex in the gripping state from the subunit structures of SMC1 (full length), SMC3 (full length), STAG1 (HEAT repeats domain, Residues 86-1052), NIPBL (HEAT repeats domain, Residues 1193-2628) and RAD21 (full length). To obtain the gripping state structure, we performed steered MD simulations to assemble (i). RAD21, and (ii). [NIPBL and STAG1] to a SMC1/SMC3 dimer structure in consecutive simulations (Methods). Subunits were driven to their bound state by artificial harmonic bonds created based on selected inter-subunit contacts in the gripping state cryo-EM structure [14]. As the structure approached the desired gripping state, the residue distances became sufficiently small for the native contact potential constructed from the cryo-EM structure to take effect. At that stage, we removed artificial harmonic bonds (S10 Fig). The DNA binding patches predicted by the simulations of each subunit were mapped on the resulting structural model of the whole human cohesin complex (Fig 5A).

Our structural model of the whole human cohesin complex showed the predicted DNA binding sites potentially involved in DNA loop extrusion (Fig 5A). Previous studies proposed various mechanisms to explain loop extrusion. One of them is the DNA segment capture model [13,32,33]. This model describes the ATP hydrolysis cycle driving loop extrusion as three key steps: 1) ATP binding to the SMC1/3 heads induces head engagement and subsequent opening of the coiled-coil arms emanating from it. 2) A DNA segment (the increment of the main extruded loop in each cycle) is spontaneously inserted between the opened arms, with the two DNA ends held by the regions around the hinge and heads (Fig 5B). 3) After ATP hydrolysis and ADP release, the coiled-coil arms zip to close again while the DNA segment held around the hinge is pumped to the head area. Consistent with this model, we

predicted strong DNA binding sites on the top surface of the SMC1/3 head dimer (C2, C3 in Table 1) and the hinge dimer (F1 in Table 1).

NIPBL binds to and quickly dissociates from the cohesin complex during loop extrusion and is proposed to stimulate ATP hydrolysis [11,34]. Therefore, NIPBL is likely involved in step 2, after head engagement and before the ATP hydrolysis happens, with the spatial organization of SMC1, SMC3, NIPBL, STAG1, and DNA as in our whole complex based on the cryo-EM structure [14]. In this stage, the distance between the hinge dimer and head dimer is around 10 nm (in the case where the coiled-coil arms of SMC1 and SMC3 are not folded, the distance is around 40 nm [35]), smaller than the persistence length of DNA of 45 nm [36]. However, the loop extrusion speed (0.5–1 kbp/s) and ATPase rate (2 ATP/s) gives a rough estimation of step size up to ∼1 kbp (340 nm) [7,8]. A more direct step size measurement of condensin, a member of the SMC complexes family with a similar size as cohesin, captures step sizes over 500 bp (170 nm) [37]. The interaction between cohesin subunits and DNA held by the SMC1/3 head dimer in step 2 may help to rationalize this large step size through the bending of DNA at the top of the SMC head dimer, which would allow DNA to deeply insert into the coiled-coil ring, resulting in a more extended DNA segment between hinge and heads (Fig 5B bottom).

Our structural model showed a group of DNA binding patches near SMC1/3 heads (C2, C3, H2, G1, G3 in Table 1). This group encircled the space where DNA was observed in the gripping state. The potential role of this patch group in step 2 of the DNA segment capture model is holding DNA on the top surface of SMC1/3 heads while bending DNA to favor the stabilization of a more extended DNA segment captured between the hinge and heads. The cryo-EM structure of the cohesin pentamer, including SMC1, SMC3, RAD21, and NIPBL, in the gripping state [14] identified DNA binding sites that largely overlapped with our predictions based on the simulations of the individual subunits. Interestingly, our simulations predicted one more site on STAG1 (G3). In the cryo-EM structure, the segment of DNA expected to bind to this site cannot be resolved, possibly due to structural flexibility around this region and the need to average over many complexes found in the same conformation in the cryo-EM structure.

We therefore sought to confirm the potential role of the complete patch group, including G3 on STAG1, in DNA loop extrusion. To this aim, we performed MD simulations of the cohesin complex consisting of SMC1/3 heads and emanating partial coiled-coil arms, STAG1, and NIPBL with a 78 bp DNA fragment [the same length and sequence as in the cryo-EM structure [14]] manually placed as in the cryo-EM structure (in which only 17 bp out of the 72 bp can be resolved, Fig 5C, top). In the simulation trajectories, we observed that DNA sharply bends as it reaches equilibrium (Fig 5C top, S2 Movie). We measured the angle of DNA bending projected on the plane normal to the plane spanned by SMC1/3 coiled-coil arms and aligned along the bisector defined by the coiled-coil axis (Figs 5D and S10 and Methods for details). The mean angle of DNA bending was 31.0°, with a standard deviation 14.5°. We also performed the simulations without STAG1 and NIPBL (Fig 5C bottom) and found that the mean angle was reduced to 3.1°, with a standard deviation of 7.6°. The two DNA bending angle distributions showed significant differences (student's t-test P-value: 0.00045) (Fig 5E). The larger angle found in the system with the full DNA-gripping patch group is expected to stabilize a larger loop captured between the heads and hinge and, therefore, a larger step size of loop extrusion. When binding to the DNA-gripping patch group, DNA stayed in contact with at least part of the patch group in all 20 simulation trajectories (Fig 5C top). On the contrary, when DNA was placed on the SMC1/3 head dimer without STAG1 and NIPBL, dissociation from the top surface was observed in 16 out of 20 trajectories (Fig 5C bottom, S2 and S3 Movies). More stable DNA binding in the presence of the patch group in the whole complex might facilitate extrusion by preventing the slippage of the captured DNA segment (which would lead to an unproductive ATP cycle).

To investigate the cooperative contribution of the predicted binding sites to affinity, we also performed MD simulations to quantify the binding strength of the group. A tetramer consisting of SMC1/3 heads and emanating partial coiled-coil arms, STAG1, and NIPBL was simulated with a 20 bp DNA molecule. Unlike previous systems A to H, which all have relatively simple, near-globular structures, the tetramer forms a tunnel structure, and the group of DNA binding patches is located on the inner sides of the tunnel. To increase the chance of DNA-protein interaction, we initially placed one DNA near the target patch group (Fig 5F). When treated as one patch, the dissociation rate constant of the group was $1.21 \times 10^{-7}$, while the strength of its component patch C2 (on the top surface of the SMC1/3 head dimer), C3 (on the top surface of the SMC1/3 head dimer), H2 (on the neck of NIPBL connecting its N-terminal handle and U-shaped hook), G1 (on the inside surface of STAG1 U-shaped hook), G3 (on the inside surface of STAG1 U-shaped hook) were $1.49 \times 10^{-5}$, $1.49 \times 10^{-5}$, $9.32 \times 10^{-7}$, $7.04 \times 10^{-6}$, and $9.37 \times 10^{-6}$, respectively. This marked decrease in the dissociation rate constant indicates the cooperative effect of multiple patches to stabilize DNA binding. Thus, these simulation results suggest that STAG1 and NIPBL facilitate loop extrusion by contributing to stable DNA binding and sharp DNA bending.

## Discussion

In this study, we employed residue-resolution MD simulations to identify strong DNA binding patches on the individual cohesin subunits, which may play an essential role in the molecular mechanisms of DNA loop extrusion. The validity of many of the identified patches was confirmed by comparison to mutagenesis experiments. We then constructed a model of the whole cohesin complex, including SMC1, SMC3, STAG1, NIPBL, and RAD21, which suggests how the DNA binding patches may cooperate during extrusion. Our findings provide vital clues for how the proposed DNA segment capture model of loop extrusion may be implemented in practice.

In particular, we used AlphaFold and steered MD simulations to model the ATP-bound state where SMC1/3 head domains are engaged and DNA is gripped at the top surface of the SMC1/3 head dimer, which most likely represents the state in which a DNA segment is captured by the ring compartment formed by SMC1/3 coiled-coil arms. In this state, we discovered the significant cooperative effect of multiple DNA binding patches in stabilizing and sharply kinking DNA near the top surface of the SMC1/3 head dimer, possibly facilitating the capture of a large loop (larger than the cohesin itself) to be extruded throughout the ATP hydrolysis cycle. Our study also revealed the importance of accessory protein subunits STAG1 and NIPBL. In their absence, DNA is unstable and likely to dissociate from cohesin, inhibiting productive translocation.

Although we included full-length RAD21 in our model of the whole cohesin complex, we note that RAD21 has long intrinsically disordered regions, which were not observed by cryo-EM experiments. The path of RAD21 IDR in our model (Fig 5A green) is one possibility and should not be taken as definite. A few structures with different possible RAD21 pathways are shown in S12 Fig. On the other hand, structures of SMC1, SMC3, STAG1, and NIPBL in the gripping state can be considered deterministic. With RAD21 topologically closing the closed ring structure, this model can be used as a starting point for studying the behavior of the cohesin complex following the DNA segment capture step. When the SMC1/3 head disengages, RAD21 IDR should move with considerable flexibility. At the same time, its N-terminus and C-terminus stay bound to SMC3 and SMC1, respectively. RAD21 is expected to restrict the DNA loop extruded by cohesin, preventing loop dissolving when SMC1/3 heads disengage. While RAD21 was not considered in our analysis of the DNA binding patches

due to its highly disordered nature, future studies should address how RAD21 may be able to modulate DNA binding to the whole cohesin complex.

According to the DNA segment capture model, after SMC1/3 head disengagement, coiled coils of SMC1/3 should "zip up" from hinge to head, pushing the DNA segment captured between hinge and head into the head area, resulting in the expansion of the DNA loop captured in the SMC1/3 head dimer-RAD21 compartment. This process can be studied using our model as a starting point, using longer DNA (larger than 1 kbp). However, to simulate the closing process of SMC1/3 coiled coils, experimental information on interactions between SMC1/3 coiled coils in their closed state (also known as APO state or I shape state) is required. Such data exist for yeast cohesin but not human cohesin, which is considered in the current work.

Although we have discussed the potential roles of the identified DNA binding patches in the DNA segment capture model, our study does not rule out the possibility of other loop extrusion models. The swing-and-clamp model [9] proposes that DNA translocation is realized in two steps: 1) the SMC1/3 coiled coils extend, and the hinge dimer binds to downstream DNA. 2) Coiled coils fold, bringing the hinge dimer close to the head dimer and handing over downstream DNA from the hinge domain to the head region. Our study identified a strong DNA binding patch at the hinge dimer (F1 in Table 1), indicating that it can capture DNA. The swing-and-clamp model also suggests that NIPBL can dissociate from the head region during step 1, bind to the hinge domain, and facilitate searching for downstream DNA [the NIPBL-hinge proximity is indicated by Förster resonance energy transfer (FRET) experiments [9], while NIPBL searching and binding downstream DNA remains without direct experimental evidence]. Our study identified a strong patch at the "nose" region of NIPBL (H1 in Table 1). This patch has one order of magnitude stronger affinity than the patch on the hinge dimer and two orders of magnitude stronger affinity than the patches on the top surface of the head dimer. In the DNA gripping state, this strong patch is covered due to NIPBL-STAG1 binding, but when NIPBL dissociates from cohesin [a behavior confirmed by *in vitro* [7] and *in vivo* experiments [11]], this patch should be considered a candidate for the search of free DNA in step 2. Our study adds some credibility to the hypothesis made in the swing-and-clamp model and provides a possible mutation site for future experiments to verify this hypothesis.

Finally, several loop extrusion models assume the existence of a DNA anchor site to explain how DNA translocation may be converted into loop extrusion. Accordingly, one end of the DNA loop is translocated through mechanisms such as DNA segment capture or swing-and-clamp, while the other end remains anchored at a specific cohesin site. This anchor may be realized by DNA binding or encircling by RAD21 and preventing it from diffusing. The strong DNA binding patch we identified on the emanating coiled coils from the SMC1 head domain (A1) is also a candidate site for the DNA anchor.

## Materials and methods

### Modeling of the cohesin subunits and the whole cohesin complex

AlphaFold 2 [31] was employed to create the full-length model of the cohesin subunits SMC1 and SMC3 (see below). These models were later used as a starting point for steered MD simulations to model the complete cohesin complex. AlphaFold 2 tends to favor compact, globular structures even when predicting the long antiparallel coiled-coil structures in SMC proteins (AlphaFold Protein Structure Database entry: Q14683, Q86VX4). To generate a model with realistic extended coiled coils, we predicted, using AlphaFold Colab [31], only the parts of coiled-coil structures missing from the cryo-EM structure [14]. The amino acid sequence of SMC1: residues 203-490, 656-1030, and SMC3: residues 243-492, 684-926 was used as input.

This yields reasonable structures as evaluated by pLDDT scores and compared with available crystal structures of their homologs (budding yeast SMC1 and SMC3, PDB ID: 7OGT, S13 Fig). Non-helical regions at the middle part of coiled coils in 7OGT were defined as "elbows" where coiled coils fold. Elbow position in human SMC1 and SMC3 can be predicted by finding the homologous sequences of yeast elbow. Sequences of SMC1 and SMC3 of human, mouse, budding yeast, and fission yeast were aligned using the multiple sequence alignment server T-coffee [38]. AlphaFold predicts the same positions as the folding point of coiled coils.

We used MODELLER [39] to construct a full-length molecule of SMC1 using head and hinge domain structures from the cryo-EM structure (PDB ID: 6WG3) [14] and AlphaFold modeled coiled-coil structures as templates. A short part of the coiled coil connected to the head domain is observed in 6WG3, which also exists in the coiled coils modeled by Alpha-Fold. This overlap prevented unnatural angles when MODELLER tried to connect two parts. In the hinge domain observed in 6WG3, the N terminus of the available structure starts with an alpha helix, continuous with an upstream alpha helix in the coiled-coil region. Using this as a restraint, MODELLER connects the coiled coils with the hinge domain without producing unnatural angles. The resulting full-length SMC1 model has the all-atom resolution.

Similar to SMC1, the cryo-EM structure of SMC3 also captures a short coiled-coil part connected to the head domain and an alpha helix in the hinge domain continuous with the coiled coils [14]. Using restraints similar to those of SMC1, the full-length SMC3 was modeled using MODELLER.

We modeled the structured regions of STAG1 (residues 85-1052) and NIPBL (residues 1130-2628) using MODELLER, which generated flexible structures to complete the loops missing in 6WG3.

We used MODELLER to generate 5 structures for each protein or protein domain, selected the model with the lowest value of the MODELLER objective function (molpdf) as the best model. The MODELLER consistently predicted the position of secondary structures and loops.

## DNA binding simulation assays

Each target system (SMC1 head domain; SMC3 head domain; SMC1/3 head dimer; SMC1 hinge domain; SMC3 hinge domain; SMC1/3 hinge dimer; STAG1; NIPBL; DNA gripping tetramer containing SMC1 head domain, SMC3 head domain, STAG1 and NIPBL) was separately simulated with short duplex DNA fragments. For SMC1 and SMC3, we simulated a stand-alone head or hinge domain with DNA instead of the full-length molecule. We also simulated the head dimer and the hinge dimer as their bound state in 6WG3 [14].

Each simulation contained B-type DNA fragments with the same sequence, ATAGTGAT-TATGAAAACTTT [a randomly generated 20 bp sequence with 20% GC content, same as the DNA sequence used in cryo-EM [14]]. For small systems (SMC1 hinge domain; SMC3 hinge domain) one DNA fragment was used. For larger systems (SMC1 head domain; SMC3 head domain; SMC1/3 head dimer; STAG1; NIPBL), three DNA fragments were used to increase binding site search efficiency. For the DNA-gripping tetramer, one DNA fragment was manually placed near the DNA-gripping patch group to promote efficient DNA-protein interaction. The B-DNA structure of the fragment at all-atom resolution was generated using the DNA Sequence to Structure tool [40].

A series of simulations studying the effect of different DNA sequences on affinity was performed with STAG1 and 3 B-type DNA fragments with the same sequence. In different simulation trajectories, the DNA sequence was either TCAGGTTTCACAGGGACAAA (45% GC content) or GGCCCCAAGCGCGGGTTCCC (80% GC content)

CafeMol generated the residue-resolution models of the various systems based on the AICG2+ [25] and 3SPN2.C [20] models for proteins and DNA, respectively. Each amino acid is represented by one bead centered at the $C_\alpha$ atom, while each nucleotide is represented by three beads corresponding to phosphate, sugar, and base groups. The AICG2+ model stabilizes a protein in its native state through bonded and non-bonded interactions based on the reference structure. Proteins and DNA interact via excluded volume and Debye-Hückel electrostatic interactions. Ionic strength was set at 300mM, higher than physiological conditions, to speed up the dissociation between DNA and a protein. In our simulations, the charges of surface beads in the residue-resolution protein models were optimized by RESPAC [26] to match the surface electrostatic field calculated from the respective all-atom structures using the PDB2PQR and APBS software tools [41]. The combination of AICG2+ and 3SPN2.C models has been successfully applied to study the dynamics of many protein-DNA systems [21,22,42], as it allows a significant speedup compared to all-atom models while retaining sufficient detail to reveal critical insights into molecular mechanisms.

Simulations for SMC1 head domain, SMC3 head domain, head dimer, STAG1, and NIPBL were carried out in a 200 Å each side cubic box with periodic boundary. The SMC1 hinge domain, the SMC3 hinge domain and the hinge dimer were simulated in a 100 Å each side cubic box with periodic boundary. The DNA-gripping tetramer was simulated without setting a boundary. Each system was simulated using Langevin dynamics for 20 trajectories (each $10^8$ steps of $\Delta t = 0.15\,fs$). Note that CafeMol uses a low friction parameter to speed up the dynamics, and we used an ionic strength at 300mM, higher than the physiological condition to speed up the dissociation of DNA-protein binding, so the protein-DNA interaction time scale directly obtained from the simulations does not represent the actual time scale *in vivo*.

## DNA binding-patch calling and dissociation rate constant calculation

A residue was considered in contact with a DNA fragment when the minimal distance between the residue and any atom from the DNA fragment was below a threshold of 8Å, slightly larger than the Debye length of 5.6Å). Trajectories were analyzed using the Python library MDanalysis [43].

Surface residues were clustered into binding patches according to the Jaccard distance, which describes how often residues bind to the same short DNA fragment [44] (S3 Fig):

$$d_J(i,j) = 1 - \frac{\left| Intersection\ (i,j) \right|}{\left| Union(i,j) \right|}$$

where the second term describes similarity between two sample sets, $Intersection\ (i,j)$ is the number of trajectory frames in which residues $i$ and $j$ are in contact with the same DNA molecule, and $Union(i,j)$ is the number of frames in which either residue $i$ or $j$ is in contact with any DNA fragment. Residues were hierarchically clustered [45] into binding patches using their Jaccard distance. Jaccard distance calculation and clustering were done using R [46].

After identifying binding patches, a binding event was considered a continuous series of timesteps where at least one residue in the binding patch of interest was in contact with a DNA fragment of interest. For one binding patch, the lifetime of all binding events involving each DNA fragment in all trajectories are calculated, with the events considered independent. We conducted bootstrapping [47] to estimate the value and error of the dissociation rate constants $k_{off}$. With $N_0$ being the total amount of events, sample size is set the same as $N_0$ and sample is drawn 1000 times with replacements. Mean value of $k_{off}$ is used as the strength of the patch, and standard deviation of $k_{off}$ is used as error. $k_{off}$ in each sample is calculated as follow:

The survival probability over time of binding events in each sample was calculated as

$$P(t) = \frac{N(t)}{N_0}, \text{ where } N(t) = \sum_{\tau > t} n_\tau \text{ is the number of events with a lifetime larger than } t,\text{ The}$$

survival probabilities displayed two regimes, one corresponding to transient interactions giving lifetimes typically shorter than some breakpoint $t_1$, and one corresponding to stable binding giving lifetimes larger than $t_1$ (S5 Fig). To estimate the dissociation rate $k_{off,\ stable}$, we fitted the $\ln(P(t)) \sim t \big|_{t > t_1}$ data points to a linear model. Discrete-time points $\{t_i\}$ were chosen so that $\{N(t_i)\}$ formed an arithmetic progression. In this way, data points were equally weighted when performing linear regression. The breakpoint $t_1$ was determined using the R package called "segmented" [48].

## HADDOCK

We used the all-atom resolution structures of SMC1 head domain or structured regions of NIPBL as input structure 1 in the HADDOCK 2.4 web server [30], and the all-atom resolution structure of 20 bp B-type DNA ATAGTGATTATGAAAACTTT as input structure 2.

All surface residues in the protein structure were set as "passive", and all residues in the DNA were set as "active" so that HADDOCK searches for the docking interface on the entire surface.

In each HADDOCK run, the positively charged residues in one of the patches were set as "active" to validate each specific DNA binding patch predicted by our MD simulation.

## Modeling of the whole cohesin complex in gripping state

Here, we describe the simulation protocol employed to generate the complete pentameric model of cohesin in the gripping state. We start from the full-length SMC1/3 complex generated by AlphaFold2 (S10A and S10B Fig), and we drive this toward the cryo-EM reference with PDB ID 6WG3 [14] by a combination of steered MD and switching-Gō simulation protocol [49]. In the initial structure, SMC1 and SMC3 dimerize solely at the hinge but not at the heads, while the coiled coils fold at the two elbow positions. To allow the structure to relax toward the reference gripping state, the local bond length, bond angle, dihedral angle interactions, and non-local contact interactions for the loop structures at the elbow positions of SMC1 and SMC3 were replaced by generic flexible local potentials [50]. Intra-protein native contacts for positions except the elbow were based on the initial AlphaFold models. In contrast, SMC1/3 inter-protein native contacts were solely based on the hinge dimer of the reference 6WG3 structure in the first simulation phase.

We then set up a first steered MD simulation to relax the tightly folded initial structure so that the two subunits can interact as in the target 6WG3 gripping state (S10A and S10B Fig). The SMC1/3 hinge was softly fixed by anchoring one amino acid particle $A_1$ to its initial position using a harmonic spring

$$V_1 = \begin{cases} k_a (r_{10} - l_a)^2 & (r_{10} > l_a) \\ 0 & (r_{10} < l_a) \end{cases},$$

where $r_{10}$ is the distance between the position of $A_1$ and its initial position. Two ghost particles were defined at the position of two amino acid particles $A_2$ and $A_3$ in the SMC1 and SMC3 head domains, respectively. Ghost particles were connected to corresponding amino acid particles via a harmonic spring:

$$V_i = k_g \left( r_i - r_{gi} \right)^2, \quad i = 2,3,$$

where $r_i$ is the position of $A_i$, and $r_{gi}$ is the position of the ghost particles. Ghost particles move in a fixed direction at a constant velocity, pulling the head domains away from hinge domains (S10A Fig).

$$r_g = \left( x + v_x t, y + v_y t, z + v_z t \right).$$

The steered MD simulation was performed for $10^6$ steps with $k_a = 0.1 kcal \cdot mol^{-1} \cdot \text{Å}^{-2}$, $k_g = 1.0 kcal \cdot mol^{-1} \cdot \text{Å}^{-2}$, and $v_g = 0.001 \text{Å} \cdot \left( \text{unit time} \right)^{-1}$ (unit time in CafeMol is 49fs). This is then followed by an unbiased MD simulation of $10^7$ steps to obtain relaxed structures. The steered MD and the unbiased relaxation were conducted without setting a boundary. Simulation timestep and protein-protein interaction force-field were set as in the previous "DNA binding simulation assays" section.

Starting from the output of the previous simulations, we then proceeded to construct a model of the SMC1-SMC3-RAD21 trimer, in which SMC1 and SMC3 dimerized at the hinge, head domains were disengaged, and RAD21 connected the SMC1 head and the SMC3 head to form a closed ring (S10C and S10D Fig). We started from the extended SMC1/3 dimer structure and the structure of the relaxed full-length RAD21 generated by AlphaFold. Alpha-Fold predicted the majority of RAD21 as unstructured, with low pLDDT scores, indicating IDRs [31]. Disordered regions predicted from sequence using DISOPRED3 [51] agreed with AlphaFold2 predictions. Based on the 6WG3 target, we manually picked five pairs of inter-acting amino acid particles, two from the SMC1-RAD21 binding interface and three from the SMC3-RAD21 interface. Harmonic bonds $V = k_b \left( r_{ij} - l_{0,ij} \right)^2$ were applied to each pair, with $r_{ij}$ being the distance between two amino acids of each pair and bond length $l_{0,ij}$ set as the distance of each pair in 6WG3. The steered MD simulations were performed first for $10^6$ steps with $k_b = 0.001 kcal \cdot mol^{-1} \cdot \text{Å}^{-2}$, and then for $10^6$ steps with $k_b = 0.1 kcal \cdot mol^{-1} \cdot \text{Å}^{-2}$. This successfully generated a SMC1-SMC3-RAD21 complex with a closed ring topology.

We then constructed a model of the SMC1-SMC3-NIPBL-STAG1-RAD21 pentamer (S10E and S10F Fig). To set up the initial structure for the steered MD simulation, we initially placed the STAG1 and NIPBL subunits near the SMC1-SMC3-RAD21 trimer structure obtained in the previous step. An analogous protocol, as before, was followed to add harmonic bonds between the STAG1-RAD21 and NIPBL-RAD21 interfaces based on the target cryo-EM structure [14]. Steered MD simulations were performed for $10^6$ steps with $k = 0.001 kcal \cdot mol^{-1} \cdot \text{Å}^{-2}$, and then for $10^6$ steps with $k = 0.1 kcal \cdot mol^{-1} \cdot \text{Å}^{-2}$. RAD21 harmonic bonds from the previous step were also turned on, with $k = 0.1 kcal \cdot mol^{-1} \cdot \text{Å}^{-2}$. The Initial placement of NIPBL and STAG1 in this step determined the pathway RAD21's IDR takes from its N-terminus near the SMC3 head to its C-terminus near SMC1 head. (S12 Fig).

Finally, we constructed the SMC1-SMC3-NIPBL-STAG1-RAD21 pentamer (the whole cohesin complex) in the DNA gripping state (S10F to S10H Fig), producing the complete model consistent with the observed cryo-EM structure (PDB ID: 6WG3) [14] but including all amino acids in the complex. The same protocol for adding harmonic bonds was applied to the SMC1 head domain-SMC3 head domain, SMC1 head domain-NIPBL, SMC3 head domain-NIPBL, and NIPBL-STAG1 interfaces. Steered MD simulations were performed first for $10^6$ steps with $k = 0.001 kcal \cdot mol^{-1} \cdot \text{Å}^{-2}$, and then for $10^6$ steps with $k = 0.1 kcal \cdot mol^{-1} \cdot \text{Å}^{-2}$. Harmonic bonds from the previous step were also turned on with $k = 0.1 kcal \cdot mol^{-1} \cdot \text{Å}^{-2}$. At this stage, the protein interfaces described above were close enough for the native AICG2+ contacts based on the reference cryo-EM structure to be effective. We, therefore, switched off

the previous harmonic bonds and added the AICG2+ intra-molecule native contact interactions based on the reference gripping state structure (PDB ID: 6WG3) [14], except for those at the SMC1 hinge domain-STAG1 and SMC1 hinge domain-NIPBL interfaces. The potential-switching simulation was performed for $10^6$ steps. Next, we added harmonic bonds to SMC1 hinge domain-STAG1 and SMC1 hinge domain-NIPBL interfaces and performed steered MD simulations for $10^6$ steps with $k = 0.001 kcal \cdot mol^{-1} \cdot Å^{-2}$ and for $10^6$ steps with $k = 0.1 kcal \cdot mol^{-1} \cdot Å^{-2}$. Harmonic bonds at other interfaces were not applied here, but AICG2+ interactions maintained the protein subunit bindings. Potential-switching to the native contact interactions at the SMC1 hinge domain-STAG1 and SMC1 hinge domain-NIPBL interfaces was performed for $10^6$ steps to obtain the final structure of the whole cohesin complex in the DNA gripping state.

All simulations in this section were performed using CafeMol with the same settings described for the unbiased MD simulations.

## Modeling of cohesin tetramer containing SMC1/3 heads, STAG1 and NIPBL to study the DNA-gripping patch group

To construct the initial structures in Fig 5C and 5F, we used previously constructed models of SMC1/3 head dimer, STAG1, and NIPBL to build a tetramer containing the DNA-gripping patch group we were interested in. We added harmonic bonds to the SMC1 head domain-NIPBL, SMC3 head domain-NIPBL, and NIPBL-STAG1 interfaces following the same protocol as the previous section. The SMC1 head domain–SMC3 head domain interface was held together by the AICG2+ intra-molecule native contact interactions based on the gripping state structure.

To simulate the effect of DNA-gripping patch group on long DNA, a 3SPN.2C model of 72 bp dsDNA [sequence: TGGTTTTTATATGTTTTGTTATGTATTGTTTA TTTTCCCTTTAA-TTTTAGGATATGAAAACAAGAATTTATC, the same as in the cryo-EM experiment [14]] was manually inserted in the tunnel surrounded by patches of interest as an initial structure. To simulate the effect of SMC1/3 head-dimer as a control, we manually placed the same dsDNA near the top surface of an SMC1/3 head-dimer model. Each system was simulated for 20 trajectories ($10^7$ steps in each trajectory), with timestep and force-fields set as in the previous "DNA binding simulation assays" section.

To measure the dissociation rate constant of the DNA-gripping patch group, one 20 bp dsDNA (with the same sequence as the previous DNA dissociation simulations) was placed at two different positions near the patch group to create two different initial structures.

## Definition and statistical analysis of DNA bending angle θ

From models of the SMC1/3 head dimer or tetramer containing SMC1/3 heads, STAG1, and NIPBL (Fig 5C), we picked the particles $B_1$ (SMC1/S161) and $B_2$ (SMC3/T165) representing the base of SMC1 and SMC3 coiled coils, $E_1$ (SMC1/E202) and $E_2$ (SMC3/N243) representing the end of emanating SMC1 and SMC3 coiled coils. The center of the top surface of the SMC1/3 head dimer C was defined as the geometric center of $B_1$ and $B_2$. The plane spanned by two coiled coils was defined by vectors $V_1 = \overrightarrow{CB_1}$ and $V_2 = \overrightarrow{CB_2}$. The plane normal to this plane was defined by the vector perpendicular to the plane $x = V_1 \times V_2$ and the bisector $z = (V_1 + V_2)/2$. The DNA trajectory was projected on the plane $xz$ (Fig 5D).

From the coordinates of the DNA model projected on the plane $xz$, we selected the sugar particle of the first nucleotide $(a_1, b_1)$ and the 11th nucleotide $(a_2, b_2)$ at the 5' end of each strand of the dsDNA. Considering the B-DNA helix is about 10 base pairs per turn, vectors $V_a = \overrightarrow{a_1 a_2}$ and $V_b = \overrightarrow{b_2 b_1}$ are approximately parallel to the track of the center of the DNA

double helix. Therefore, the DNA bending angle was calculated as $\theta = \arccos\left(\dfrac{V_a \cdot V_b}{|V_a||V_b|}\right)$, while not accounting for the effect of the natural twist of DNA helix. This twist introduces complexity to the calculations, as it influences the overall shape and structure of DNA.

To confirm the simulation reached convergence, we calculated the rolling average of $\theta$ over all trajectories in a moving time window of 500 frames ($5 \times 10^5$ steps). The rolling averages showed slight fluctuation, but no upward or downward trend over time (S11B Fig). The first half of each trajectory was discarded to avoid bias induced by initial structure. The mean value of DNA bending angle $\theta_i\big|_{i=1,2,\ldots,20}$ is calculated for each trajectory. $\theta_i$ calculated from each trajectory is considered independent. The mean value of $\theta_i$ is used as DNA bending angle $\theta$ in the result section. In the case of SMC1/3 head dimer and 72 bp DNA simulation system, only the 4 trajectories in which DNA did not dissociate from protein domain throughout the simulation were used to calculate $\theta_i$.

## Supporting information

**S1 Fig. Sequence and structure alignment of HAWK subunits in human and yeast cohesin.** (A) Sequences of Homo sapiens NIPBL, Saccharomyces cerevisiae SCC2, and Schizosaccharomyces pombe MIS4 protein aligned using the T-coffee server [38]. T-coffee combines popular MSA algorithms and evaluates the alignment confidence of each column by transitive consistency score (TCS). TCS indicates the consistency between different MSA algorithms. Therefore, a high score indicates better sequence conservation. Protein structures are indicated below TCS scores, with rectangles indicating structured domains and lines indicating intrinsically disordered regions. (B) Sequences of Homo sapiens STAG1, Saccharomyces cerevisiae SCC3, and Schizosaccharomyces pombe PSC3 protein aligned using the T-coffee server.
(PDF)

**S2 Fig. DNA contact frequency of each amino acid mapped on subunit structures. Contact frequency $f$ of each amino acid residue particle is defined as $N=N/N0$, where N is number of frames the corresponding particle is in contact with any DNA segment, $N0$ is the total number of frames in all simulation trajectories.** (A) SMC1 head domain with emanating partial coiled coil arm. (B) SMC3 head domain with emanating partial coiled coil arm. (C) SMC1/3 head dimer with emanating partial coiled coil arms. (D) SMC1 hinge domain. (E) SMC3 hinge domain. (F) SMC1/3 hinge dimer. (G) the HEAT repeats domain of STAG1. (H) the HEAT repeats domain of NIPBL.
(PDF)

**S3 Fig. Illustration of DNA binding-patch calling method.** (A) definition of the DNA contact pattern similarity between two amino acid residues through Jaccard distance. (B) hierarchical clustering of surface amino acid residue particles of SMC1 head domain.
(PDF)

**S4 Fig. Identified DNA binding patches mapped on subunit structures. Patches affected by SMC1/SMC3 head dimerization and SMC1/SMC3 hinge dimerization are labeled in figures.** (A) SMC1 head domain with emanating partial coiled coil arm. (B) SMC3 head domain with emanating partial coiled coil arm. (C) SMC1/3 head dimer with emanating partial coiled coil arms. (D) SMC1 hinge domain. (E) SMC3 hinge domain. (F) SMC1/3 hinge dimer. (G) the HEAT repeats domain of STAG1. (H) the HEAT repeats domain of NIPBL.
(PDF)

**S5 Fig. Survival rate–time distribution of DNA-patch contact events.** The logarithm ratio of contact events with lifetime $\tau$ larger than time t was plotted as a function of $t$. The exponential decay phase with a smaller dissociation rate constant indicating binding events was plotted as green data points, and the linear models fitted using these data points were plotted as green lines.
(PDF)

**S6 Fig. Example time series of IDs of amino acid particles in contact with DNA.** Visualized in the same way as Fig 3B.
(PDF)

**S7 Fig. Sequence independent DNA binding affinity.** DNA binding affinity simulation assays were performed using STAG1 structured domain and 20 bp B-type DNA fragment, with CG content of 0.2, 0.5, or 0.8. The dissociation rate constants $Koff$ calculated from assays with different DNA fragments were plotted.
(PDF)

**S8 Fig. Reproducing surface electrostatic field with RESPAC.** (A) Illustration of NIPBL's "neck" position. (B) The electrostatic potential around the NIPBL HEAT repeats domain calculated and visualized using the APBS plugin [41] in PyMOL [52] (C & D) DNA contact frequency of each amino acid mapped on the coarse-grained model of NIPBL HEAT repeats domain. DNA contact frequency was calculated using simulations conducted with (C) and without (D) RESPAC charge calibration.
(PDF)

**S9 Fig. Paralogous protein-DNA complex structures and molecular docking structures as validation for novel predicted DNA binding patches.** (A) Strong DNA binding patch A1 (see Table 1, S1 Table) identified by our simulation is colored in red. By aligning the sequence of SMC1's head domain and emanating coiled-coil arms (observed part in 6WG3) with RAD50's head domain and emanating coiled-coil arms (observed part in 4W9M) with T-coffee [1], we identified patch A1's paralogous sequence in RAD50, colored in orange. (B) Molecular docking prediction of SMC1-DNA complex and NIPBL-DNA complex. The strongest DNA binding patches, A1 (SMC1) and H1 (NIPBL), predicted by MD simulations are colored in blue. (C) Molecular docking prediction when specifying positively charged residues in NIPBL's patch H2 as active sites. The patch H2 predicted by MD simulations is colored in blue.
(PDF)

**S10 Fig. Steered MD setups to construct the whole cohesin complex model.** (A~B) Steered MD simulation to obtain an SMC1-SMC3 dimer structure with open coiled-coil arms. (C~D) Steered MD simulation to obtain SMC1-SMC3-RAD21 complex. (E~F) Steered MD simulation to obtain the cohesin whole complex structure in which accessory subunits STAG1 and NIPBL are bound only to RAD21 but not SMC1 and SMC3. (F~H) Steered MD simulation and potential switching simulations to obtain cohesin whole complex structure in the ATP bound and DNA bound state in cryo-EM structure 6wg3.
(PDF)

**S11 Fig. Definition and analysis of DNA bending angle.** (A) Definition of project plane used in Fig 4D. two vectors $x,z$ define the plane cutting the SMC1-SMC3 ring compartment. $x$ is the vector normal to coiled coil arms emanating from SMC1/3 head dimer, $z$ is the bisector of the SMC1 coiled coil arm and SMC3 coiled coil arm. (B) Convergence analysis of $\theta$. Rolling average of $\theta$ in a $5 \times 10^5$ step window in all 20 trajectories (black) and mean value of $\theta$ in all time steps in all trajectories (red) are plotted.
(PDF)

**S12 Fig. Example cohesin whole complex structures with different RAD21 pathways.**
(A) The majority of RAD21 IDR is geometrically located between SMC proteins (SMC1 and SMC3) and HEAT repeat proteins (STAG1 and NIPBL). (B) The N-terminus of RAD21's IDR is on the "back" side of the complex. (C) The C-terminus of RAD21's IDR is on the "back" side of the complex.
(PDF)

**S13 Fig. Prediction of SMC1 and SMC3 missing reagions.** Positions of yeast SMC1 (A & B) and SMC3 (C & D) elbow in the crystal structures, compared with the positions of their homolog sequences in human SMC1 and SMC3 in the predicted structures.
(PDF)

**S1 Table. Surface residues of each DNA binding patch.**
(XLSX)

**S1 Movie. Representative trajectory of the DNA binding affinity simulation for System A, which consists of the SMC1 head domain along with the cryo-EM-resolved portion of the coiled-coil arms.** Gray-colored beads represent the SMC1 head domain with an extending partial coiled-coil arm. Green, red, and blue beads correspond to three distinct 20 bp DNA segments. The blue cubic box denotes the periodic boundary conditions applied in the simulation.
(ZIP)

**S2 Movie. Representative simulation trajectory of DNA (gray) being bent by the DNA-gripping patch group on a protein tetramer composed of SMC1 (cyan) and SMC3 (blue) head domains with extending partial coiled-coil arms, STAG1 (orange), and NIPBL (pink).**
(ZIP)

**S3 Movie. Representative simulation trajectory of DNA (gray) interacting with and subsequently dissociating from the DNA-binding patches located on the top surface of a protein dimer composed of SMC1 (cyan) and SMC3 (blue) head domains with extending partial coiled-coil arms.**
(ZIP)

## Acknowledgement

We would like to thank the members of the theoretical biophysics laboratory at Kyoto University for discussions and assistance throughout this work.

## Author contributions

**Conceptualization:** Chenyang Gu, Shoji Takada, Giovanni B. Brandani, Tsuyoshi Terakawa.

**Data curation:** Chenyang Gu.

**Formal analysis:** Chenyang Gu.

**Funding acquisition:** Chenyang Gu, Shoji Takada, Tsuyoshi Terakawa.

**Investigation:** Chenyang Gu.

**Methodology:** Chenyang Gu, Shoji Takada, Giovanni B. Brandani, Tsuyoshi Terakawa.

**Project administration:** Giovanni B. Brandani.

**Software:** Chenyang Gu.

**Supervision:** Shoji Takada.

**Visualization:** Chenyang Gu.

**Writing – original draft:** Chenyang Gu.

**Writing – review & editing:** Chenyang Gu, Shoji Takada, Giovanni B. Brandani, Tsuyoshi Terakawa.

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
