## [Decision Letter · Decision Letter 0]

26 Oct 2024

PCOMPBIOL-D-24-01563Molecular dynamics simulations of human cohesin subunits identify DNA binding sites and their potential roles in DNA loop extrusionPLOS Computational Biology Dear Dr. Terakawa, Thank you for submitting your manuscript to PLOS Computational Biology. After careful consideration, we feel that it has merit but does not fully meet PLOS Computational Biology's publication criteria as it currently stands. Therefore, we invite you to submit a revised version of the manuscript that addresses the points raised during the review process. Please submit your revised manuscript within 60 days Dec 26 2024 11:59PM. If you will need more time than this to complete your revisions, please reply to this message or contact the journal office at ploscompbiol@plos.org. Please include the following items when submitting your revised manuscript: * A rebuttal letter that responds to each point raised by the editor and reviewer(s). You should upload this letter as a separate file labeled 'Response to Reviewers'. This file does not need to include responses to formatting updates and technical items listed in the 'Journal Requirements' section below.* A marked-up copy of your manuscript that highlights changes made to the original version. You should upload this as a separate file labeled 'Revised Manuscript with Track Changes'.* An unmarked version of your revised paper without tracked changes. You should upload this as a separate file labeled 'Manuscript'. If you would like to make changes to your financial disclosure, competing interests statement, or data availability statement, please make these updates within the submission form at the time of resubmission. Guidelines for resubmitting your figure files are available below the reviewer comments at the end of this letter. We look forward to receiving your revised manuscript. Kind regards, Benjamin Hall, DPhilAcademic EditorPLOS Computational Biology Arne ElofssonSection EditorPLOS Computational Biology Feilim Mac GabhannEditor-in-ChiefPLOS Computational Biology Jason PapinEditor-in-ChiefPLOS Computational Biology  **Journal Requirements:** **Additional Editor Comments (if provided):****Reviewers' comments:** Reviewer's Responses to Questions

**Comments to the Authors:**

Reviewer #1: The manuscript details a considerable body of complex computational work aimed at shedding more light on the structure, function and mechanisms underpinning the activity of cohesin. The approach appears to be a development of the studies on condensin that the team have previously published in PLoS Computational Biology (reference 21). The work is well described, appears to be of high quality, and provides intriguing and experimentally testable ideas about the roles of various potential DNA-binding patches on the complex that contributes to ongoing debate in the area over competing mechanistic hypotheses.

My only minor reservations relate to a general lack of the application of statistical methods to quantify the precision/significance of observations that are made.

So for example, what are the error bars on the dissociation rate constants (and how are they calculated)? What is the evidence that the bending angle investigations (Figure 5) are converged (where do figures like "32.4°±0.3°" on page 8 come from - the data in Figure 5E suggests much greater flexibility and lower if any statistical significance between the behaviour of dimer and tetramer systems).

More qualitatively, it would be good to have some commentary on how deterministic the large-scale model building work has been. Both MODELLER and steered MD simulation methods can potentially generate significantly different predictions (maybe) from run to run - what sort of variablity was observed here - what structural features can be regarded as "well resolved" and which should be regarded as more approximate?

Reviewer #2: Reviewer Expertise: Molecular Dynamics, Protein Structure

Gu et al. present a molecular dynamics study of human cohesion subunits and identify DNA binding sites, suggesting functional roles for these in DNA loop extrusion. The study involves the use of reduced representation molecular dynamics simulations (including steered MD) to simulate DNA-protein interactions for a variety of cohesion models (single subunits and dimers), and identifies proposed binding sites for DNA.

The introduction is well written and comprehensive – it lays out well the previous work in the field, and highlights the proposed mechanisms for cohesion DNA binding. The methods are well written and also comprehensive, and the authors use a blend of Alphafold predicted structures with MODELLER-based homology modelling to generate a comprehensive library of structures. These structures are then subjected to coarse grained molecular dynamics simulations to study DNA binding and conformation. As far as I can see the MD methods employed are appropriate, though I will note that I am not an expert in this particular type of coarse grained MD. The figures are generally of good quality and well described and the findings consistent.

An overarching comment I would make is that whilst the simulations are employed appropriately, a single method is used throughout the study, and whilst the authors suggest potential mutations that could be validated experimentally, they could also do some level of validation themselves using other techniques such as all-atom molecular dynamics (AT-MD) or docking/binding calculations using eg. FoldX based on frames from their simulations.

Comments:

- Standard of English could use some slight improvement. The manuscript is fully readable in current state, but grammar is not fully correct. I would recommend having the manuscript proofed.

- Is the DNA sequence important for binding? The authors have only simulated one sequence (reasonably – as simulating many would exponentially increase the simulation time required), is this likely to be meaningful? If there is a specific sequence/CG rich region that binds, and if this happens to be eg. On the end of DNA sequence, will this not bias the simulations somewhat?

- Have the authors considering performing preliminary validation themselves? I could imagine short scale all atomistic (AT) MD simulations, or energy calculations such as FoldX could be used to induce proposed mutations and calculate DNA binding affinity differences? I would expect some sort of validation of this type (eg. Multiscale MD) to be performed for publication in this journal.

- Leading on from the previous comment – the authors suggest amino acids found to be critical experimentally and that overlap with their binding patches. Are there other critical residues that are not overlapping? I.e. have the authors found that a majority of the critical residues are in their patches, or have they cherry picked a subset that happen to fall into their binding sites? Statistics could help here, or simulations/small scale docking studies of some of the mutations to show they are consistent.

- Statistics could be performed to prove the difference between DNA bending angles in the different systems (Figure 5E). The difference looks significant to me but I would suggest calculating it and presenting it to bolster your claim that it is different.

- The remarks under Table 1 are not numbered in the document I received (may be a rendering issue in the journal format).

Overall this is a technically well performed study, that suggests answers to a relevant question. Methods applied are appropriate, and analysis is performed correctly (to my knowledge/expertise). I feel however that more validation in silico could be performed by the authors – eg. Docking/binding affinity calculations of mutations, small scale AT MD simulations.

Reviewer #3: The submitted manuscript studies the mechanism of DNA loop extrusion by the SMC (structural maintenance of chromosomes) complex,particularly focusing on cohesion. Using molecular dynamics simulations with AICG2+ and 3DNA models , the authors look to identigy DNA-binding sites across cohesin subunits (SMC1, SMC3, STAG1, NIPBL) and assess their role in facilitating DNA loop extrusion. DNA-binding patches were mapped to the proteins, their binding affinities were computed and finally models were constructed to explain how these interactions might drive the loop extrusion process. The study provides insights into possible ways of DNA handover during loop extrusion and highlights the cooperative roles of accessory subunits such as STAG1 and NIPBL.

The manuscript is well presented although in some cases it could benefit of a bit more rigorous details:

For instance, how many initial modeller structures were used? Does the initial shape of the modelled loop affect your simulations? Please do comment on that. Perhaps it would be useful to comment how the induced fitting is modelled with the proposed methodology for the untrained reader.

A point of concern to me is, if the choice of modelling only 1 random sequence in the small DNA duplexes binding essay affect the assignation of binding patches in the protein. Perhaps several randomly generated sequences would be useful to avoid any bias and increase statistical significance. Are they replicas or just one single simulation run? This should be addressed. How does this model compare to traditional flexible docking approaches to recognise the binding sites?

Minor comment:

The remarks labels in Table 1 are not shown (maybe got lost after pdf creation?)

Overall, the manuscript presents a reasonable mechanism for DNA loop extrusion mediated by cohesins, using a simulation approach. The findings offer valuable predictions for future experimental research and expand the knowledge on how cohesins may interact with DNA during loop extrusion. It could use some improvements in clarity and discussions on experimental implications – and lack thereof- This study has the potential to be a significant addition to the field.

**Have the authors made all data and (if applicable) computational code underlying the findings in their manuscript fully available?**

Reviewer #1: **No: ** The URL https://doi.org/10.51093/bsm-00070 does not appear to be valid

Reviewer #2: Yes

Reviewer #3: **No: ** There is a link with an associated doi in the Data availability section of the manuscript but the link did not work today.

PLOS authors have the option to publish the peer review history of their article (what does this mean? ). If published, this will include your full peer review and any attached files.

**Do you want your identity to be public for this peer review?** For information about this choice, including consent withdrawal, please see our Privacy Policy .

Reviewer #1: No

Reviewer #2: **Yes: ** David Shorthouse

Reviewer #3: No

 **Figure resubmission:**While revising your submission, please upload your figure files to the Preflight Analysis and Conversion Engine (PACE) digital diagnostic tool, https://pacev2.apexcovantage.com/. PACE helps ensure that figures meet PLOS requirements. To use PACE, you must first register as a user. Registration is free. Then, login and navigate to the UPLOAD tab, where you will find detailed instructions on how to use the tool. If you encounter any issues or have any questions when using PACE, please email PLOS at figures@plos.org. Please note that Supporting Information files do not need this step. If there are other versions of figure files still present in your submission file inventory at resubmission, please replace them with the PACE-processed versions. 
---

## [Decision Letter · Decision Letter 1]

26 Jan 2025

PCOMPBIOL-D-24-01563R1

Molecular dynamics simulations of human cohesin subunits identify DNA binding sites and their potential roles in DNA loop extrusion

PLOS Computational Biology

Dear Dr. Terakawa,

Thank you for submitting your manuscript to PLOS Computational Biology. After careful consideration, we feel that it has merit but does not fully meet PLOS Computational Biology's publication criteria as it currently stands. Therefore, we invite you to submit a revised version of the manuscript that addresses the points raised during the review process.

Please submit your revised manuscript within 30 days Mar 28 2025 11:59PM. If you will need more time than this to complete your revisions, please reply to this message or contact the journal office at ploscompbiol@plos.org. Please include the following items when submitting your revised manuscript:

We look forward to receiving your revised manuscript.

Kind regards,

Benjamin Hall, DPhil

Academic Editor

PLOS Computational Biology

Arne Elofsson

Section Editor

PLOS Computational Biology

**Additional Editor Comments :**

Please address the minor text alterations requested by reviewer 3 for acceptance.

**Journal Requirements:**

1) Please upload a copy of Figure (9F to H) which you refer to in your text on page 16. Or, if the figure is no longer to be included as part of the submission, please remove all reference to it within the text.

2) Thank you for stating that "The data that support the findings of this study are openly available in The Biological Structure Model Archive (BSM-Arc). at https://doi.org/10.51093/bsm-00070 This link reaches a DOI Not Found page. Please amend this to a working link or provide further details to locate the data.

**Reviewers' comments:**

Reviewer's Responses to Questions

Reviewer #1: The authors appear to have responded very positively to the comments I made in my original review, I have no further concerns.

Reviewer #2: The authors have made a good faith attempt to address comments from all reviewers, and have included significantly more analysis and work to address my and the other reviewers comments. In particular the inclusion of further docking to validate findings from the simulations, and addition of statistics has significantly improved the validity of the manuscript.

Readability of the manuscript has improved, and I think it is a high quality piece of research. I thank the authors for their responses.

Reviewer #3: Many aspects have been clarified in this version, however I still have some comments:

When mentioning the choice of the best models from MODELLER, the name of the scoring method is missing.

Then new docking assay with HADDOCK was done with the structures of SMC1 head domain, whereas the new binding assays with different GC content were done on the STAG1 subunit. What is the rationale behind the choice of different cohesin subunits?

Regarding the bias on DNA sequence, the authors should report the new DNA sequences with different GC content that were used too. Is there a reason for choosing to only run it for STAG1 and not all the other subunits? This should be clarified - ideally, it would be good to have it for all subunits for completeness.

**Have the authors made all data and (if applicable) computational code underlying the findings in their manuscript fully available?**

Reviewer #1: Yes

Reviewer #2: Yes

Reviewer #3: Yes

PLOS authors have the option to publish the peer review history of their article (what does this mean? ). If published, this will include your full peer review and any attached files.

**Do you want your identity to be public for this peer review?** For information about this choice, including consent withdrawal, please see our Privacy Policy .

Reviewer #1: No

Reviewer #2: **Yes: ** David Shorthouse

Reviewer #3: No

**Figure resubmission:**
---

## [Editor Report · Decision Letter 2]

19 Feb 2025

Dear Dr. Terakawa,

We are pleased to inform you that your manuscript 'Molecular dynamics simulations of human cohesin subunits identify DNA binding sites and their potential roles in DNA loop extrusion' has been provisionally accepted for publication in PLOS Computational Biology.

Best regards,

Benjamin Hall, DPhil

Academic Editor

PLOS Computational Biology

Arne Elofsson

Section Editor

PLOS Computational Biology

---

## [Editor Report · Acceptance letter]

PCOMPBIOL-D-24-01563R2

Molecular dynamics simulations of human cohesin subunits identify DNA binding sites and their potential roles in DNA loop extrusion

Dear Dr Terakawa,

I am pleased to inform you that your manuscript has been formally accepted for publication in PLOS Computational Biology. Your manuscript is now with our production department and you will be notified of the publication date in due course.

With kind regards,

Anita Estes
